# Sensorless Posture Detection of Reluctance Spherical Motor Based on Mutual Inductance Voltage

**Jiazi Xu [1,2,3], Qunjing Wang [2,4,*], Guoli Li [2,5], Rui Zhou [2,*], Yan Wen [6], Lufeng Ju [2,3] and Sili Zhou [2,4]**

1 School of Electronics and Information Engineering, Anhui University, Hefei 230601, China; xjz303@ahu.edu.cn
2 School of Electrical Engineering and Automation, Anhui University, Hefei 230601, China; liguoli@ahu.edu.cn (G.L.); julf@ahu.edu.cn (L.J.); szhou@stu.ahu.edu.cn (S.Z.)
3 National Engineering Laboratory of Energy-Saving Motor & Control Technology, Anhui University, Hefei 230601, China
4 Collaborative Innovation Centre of Industrial Energy-Saving and Power Quality Control, Anhui University, Hefei 230601, China
5 Anhui Key Laboratory of Industrial Energy-Saving and Safety, Anhui University, Hefei 230601, China
6 School of Internet, Anhui University, Hefei 230601, China; wenyanchn@ahu.edu.cn
* Correspondence: wangqunjing@ahu.edu.cn (Q.W.); 11007@ahu.edu.cn (R.Z.)

**Abstract:** In this paper, a sensorless rotor posture detection method based on the mutual inductance voltage of the stator coil is proposed to simplify the position detection element of a reluctance spherical motor. Firstly, the numerical relationship between the stator/rotor pole misalignment angle and the mutual inductance voltage of the stator coil is analyzed, which is used as the basis for judging the spatial position of the rotor. Secondly, an experimental platform is designed to verify the consistency between the calculated value and the experimental value of the mutual inductance voltage and to determine the appropriate excitation signal. Thirdly, based on the real-time voltages generated by the stator coil mutual inductance, an intelligent algorithm is used to invert the 3-DoF (degree-of-freedom) position angle of the spherical rotor combined with the motor structure constraints. The experimental results show that the detection method has a good on-line detection effect, and the population standard deviation is within 1.8° Therefore, the developed technique can be used for replacing the position detection method with sensors.

**Keywords:** spherical motor; attitude detection; switched reluctance; mutual inductance voltage; particle swarm optimization

## 1. Introduction

Under the development of industrial automation demand for a multi-degree-of-freedom (DoF) motion mechanism, a motion mode for motors has been developed from simple one-dimensional rotation to a multi-degree-of-freedom motion. Due to the compact structure, no accumulation of motion bias, and a flexible motion mode [1], the spherical motor is especially suitable for installation and narrow motion space and has a wide application prospect in robot arms, joints, and other fields. However, due to its 3-DoF motion characteristics, the spherical motor cannot be driven in an open-loop manner. Whatever the motion is should be based on accurate attitude detection. Until now, scholars from all over the world have made extensive explorations in the position detection and drive control of spherical motors and have achieved outstanding results. However, the reported research still has some defects in position detection, especially in sensorless detection. As a result, the volume advantage of a spherical motor over a multi-DoF motion mechanism formed by a combination of multiple single-DoF motors is completely offset by a huge and complex detection mechanism, which has since become a great obstacle for spherical motors to enter actual industrial applications.

According to whether the detection device is in contact with the rotor of the spherical motor, the rotor position detection method can be divided into contact detection method and non-contact detection method. All the contact detection methods need to add a mechanical detection mechanism to the rotor, driving the encoder through the contact structure, such as the sensing systems described in [2–4]. Although the detection mechanism adopted has good detection accuracy, it occupies too much space and interferes with the rotor motion. In order to avoid the defects of the contact detection method, the works in [5–7] use the Hall element for non-contact position detection in their research process for spherical motors, locating the rotor by detecting the magnetic field of the permanent magnet in the rotor. The works in [8,9] adopted a pseudo-random array color checkered pattern coding method, through the color information detected by optical sensors, cooperating with known rotor coating code and detecting the rotor position by Machine vision. In the sensorless position detection of spherical motors, Bai uses the induced voltage generated in the stator coils of the permanent magnet spherical motor rotor flux to first realize the sensorless detection of the angular velocity and position increment of the spherical motor rotor as described in [10,11]. Compared with the results from a gyroscope, it can achieve the same accuracy and a better signal-to-noise ratio, but the detection accuracy is lower at low speed or zero-speed since the reverse voltage is difficult to detect.

In order to realize sensorless position detection of the spherical motor, a method is proposed in this paper to judge the 3-DoF position of the switched reluctance rotor by detecting the mutual inductance voltage of the stator coils. The numerical relationship between the misalignment of stator/rotor poles and the mutual inductance of coil is analyzed as the basic information to judge the spatial position of the rotor. In the process of on-line rotor position detection, according to the mutual inductance voltage information collected in real time, an intelligent optimization algorithm is used, combined with the mutual inductance voltage distribution pattern and constraints of the rotor structure to inverse the 3-Dof position angle of the rotor. Taking a 24/6 pole switched reluctance spherical motor as an experimental object as shown in Figure 1, the proposed sensorless position detection method is verified by comparing the results with the contact sensors at artificially preset posture points. The results prove the effectiveness of the method.

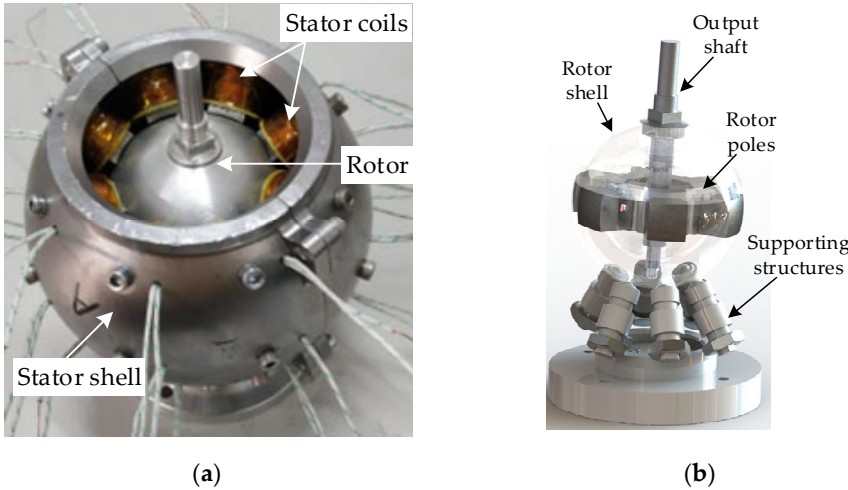

(**a**)　　　　　　　　　　　　　　　　　　　　　(**b**)

**Figure 1.** (**a**) Reluctance spherical motor, (**b**) structure of rotor.

## 2. Basic Principle of Detecting Rotation Angle from Mutual Inductance Voltage

### 2.1. Mutual Inductance Voltage Characteristics of Switched Reluctance Motor

Due to the doubly salient structure of a switched reluctance motor, the mutual inductance among windings changes with the rotor position angle. When coil A of the

reluctance motor is energized with AC, the mutual inductance voltage would be generated in an unenergized open coil B:

$$u_B = e_B = -\frac{d\Psi_{AB}}{dt} \tag{1}$$

$$\Psi_{AB} = L_{AB} i_A \tag{2}$$

where $\Psi_{AB}$ is the mutual inductance flux linkage between coil A and B, and $L_{AB}$ is the mutual inductance between coil A and B. Substituting (2) into (1) obtains

$$u_B = -L_{AB}\frac{di_A}{dt} - i_A\frac{dL_{AB}}{dt} = -L_{AB}\frac{di_A}{dt} - \omega i_A\frac{dL_{AB}}{d\theta} \tag{3}$$

where $\omega$ is the rotor angular velocity. For the spherical motor which runs in a repeated start–stop or ultra-low speed state, it can be considered that $\omega = 0$. According to the voltage equation of the switched reluctance motor [12–16], considering the input voltage of phase A winding, the following can be obtained

$$u_A = r_A i_A - e_A = r_A i_A + \frac{d\Psi_A}{dt} = r_A i_A + L_A\frac{di_A}{dt} + \omega i_A\frac{dL_A}{d\theta} \tag{4}$$

Then,

$$\frac{di_A}{dt} = \frac{1}{L_A}(u_A - r_A i_A) \tag{5}$$

Substituting (5) into (3) obtains

$$u_B = -\frac{L_{AB}}{L_A}(u_A - r_A i_A) \tag{6}$$

Equation (6) indicates that, under the condition of a stable excitation phase voltage, the relative position between stator and rotor salient poles can be indirectly judged through detecting the mutual inductance voltage values of other windings. Therefore, stator coils can be used instead of position sensors to measure the spatial position of the rotor.

### 2.2. Numerical Relationship between Mutual Inductance Voltage and Single-DoF Position Angle

The reluctance spherical motor studied in this paper adopts a structure with 24 stator poles arranged in three layers and six rotor poles distributed in one single layer, as shown in Figure 2. The stator cores are made of DT4 iron. The stator poles of the middle layer are on the spherical equator, and the stator poles of the upper and lower layers are, respectively, located on the 33° north and south latitude lines. Eight magnetic poles of each layer are distributed at equal intervals of 45° longitude. A centralized stator winding with 350 turns is placed on the stator pole. The length of unilateral air gap between stator and rotor is 1mm; the polar surface is spherical, and the size is the same. The pitch range of the rotor is 0°–33°, and the actual maximum pitch angle is 30° due to the influence of the diameter of the output shaft.

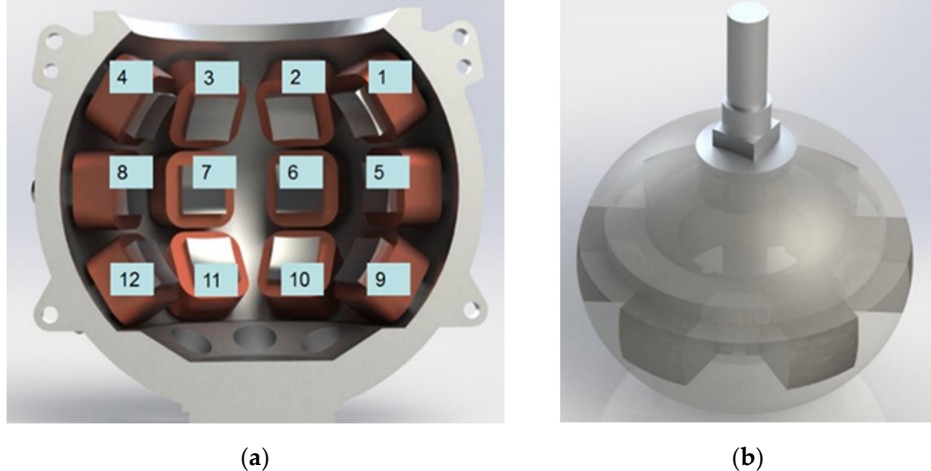

(**a**)                                                                    (**b**)

**Figure 2.** (**a**) Stator structure of 24/6 switched reluctance spherical motor, (**b**) rotor structure.

In order to obtain the numerical relationship between stator and rotor magnetic pole misalignment angle and the mutual inductance voltage, a test experimental platform is designed in this paper as shown in Figure 3. It includes the reluctance spherical motor to be tested, a contact position detection bench, an excitation signal source, a relay group, and the mutual inductance voltage detection circuit. The main parameters of the reluctance spherical motor to be tested are as mentioned above. Absolute optical encoders are used in the contact position detection bench to record the rotation angle of each degree of freedom. When the signal resource used to generate a sinusoidal signal is injected into a stator coil, the mutual inductance voltage would be generated in other related coils. The mutual inductance voltage measurement circuit is used for detecting the amplitude of the mutual inductance voltage generated in each coil. Which specific coil the signal is injected into and which coil the mutual inductance voltage is detected in might be switched by relay group.

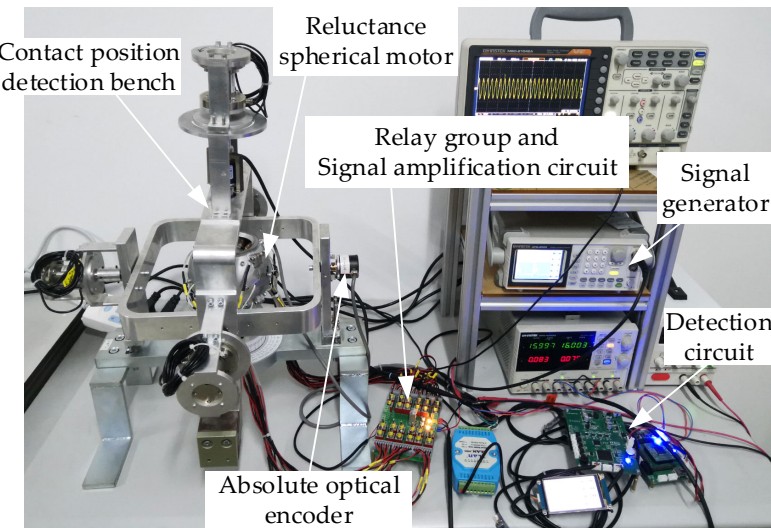

**Figure 3.** Mutual inductance voltage experimental platform.

The block diagram of the experimental platform is shown in Figure 4. The excitation coil and response coil are two of the spherical motor coils; the others are not illustrated. Due to the signal generator load limitation, in order to stabilize the amplitude of the

excitation signal when the rotor rotates, a signal amplification circuit is installed to amplify the sinusoidal signal generated by the signal generator. The signal source $IN$ comes from a function signal generator, through an amplification circuit composed of a power operational amplifier and is output as an excitation signal $u_A$ to a stator coil of the spherical motor.

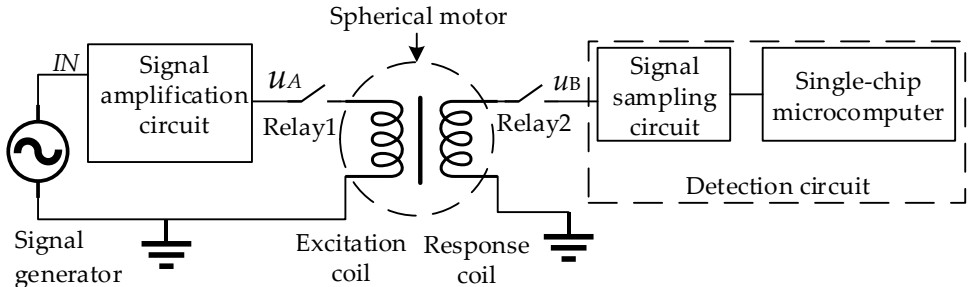

**Figure 4.** The block diagram of the experimental platform.

The excitation signal $u_A$ used in the experiment is a sine wave signal with a frequency of 50 Hz, and the mutual inductance voltage $u_B$ is also a sine wave signal with the same frequency. $u_B$ will be sampled by a signal sampling circuit. The sampling results will be converted into digital data and sent to a single-chip microcomputer. The single-chip microcomputer will process this digital data and give the amplitude of $u_B$.

In order to conveniently judge the starting point position of the misalignment angle between the stator and rotor magnetic poles, in accordance with generating the same torque, 24 stator coils are divided into 12 groups according to the spherical symmetry position. The code of each group is arranged in Figure 1a. A coil of one group located in the Eastern Hemisphere is used to inject excitation coils for $u_A$, and the mutual inductance voltage $u_B$ will be generated by the other coil of this group located in the Western Hemisphere. In this way, the mutual inductance voltage value corresponds to the angle between the axis of this coil group and the closest one of the six rotor teeth to this axis. Because of the symmetrical structure of six rotor poles, the influence of the rotor yawing on the mutual inductance voltage in each group of coils is periodic every 60°. Taking the coil group located at 22.5° east and west longitude on the spherical surface in the middle stator magnetic pole as an example, the excitation coil injects a sinusoidal AC signal with a frequency of 50 Hz and an amplitude of 0–5 V. Rotating the rotor vertically by 60° while keeping the pitch angle at 0° is one period of coil mutual inductance voltage. The finite element calculation and actual measurement are carried out on the coil mutual inductance voltage, changing with rotor position angle and excitation signal amplitude, respectively, and the results are shown in Figure 5. Among them, the red and blue waveforms shown in Figure 5a are the finite element calculation results, and the line graph shown in Figure 5b is the experimental test results. The comparison between the two results is shown in Figure 5c. It can be seen that the results measured by the experiment are highly consistent with the calculation results, and they conform to the prediction of Equation (6).

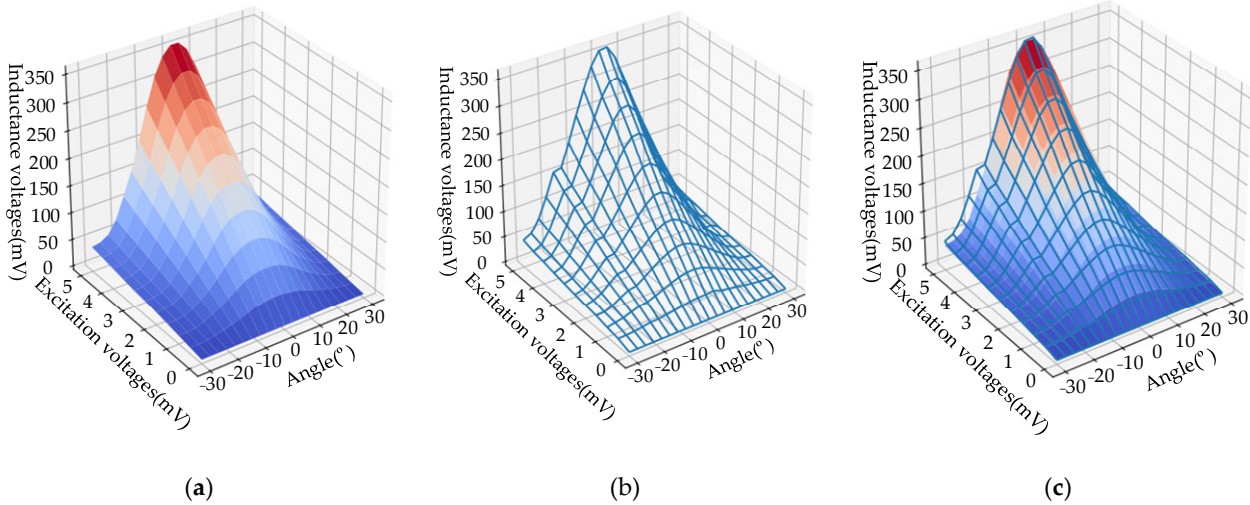

**Figure 5.** (**a**) Inductance voltages calculated by Finite Element Method (FEM), (**b**) inductance voltages measured, (**c**) comparison.

As can be seen from Figure 5, the value of the coil mutual inductance voltage reaches the maximum value at the position of zero stator and rotor poles misalignment angle and is symmetrically distributed on both sides of the peak position. According to Equation (6), as the amplitude of the input signal $u_A$ increases, the amplitude of the mutual inductance voltage $u_B$ increases. As can be seen from Figure 5, when the amplitude of the input signal $u_A$ rises from 0 V to 5 V, the mutual inductance voltage at each deviation angle position almost linearly rises accordingly, i.e., the spherical motor iron core is still in unsaturated state. Considering that the higher the input signal, the stronger the driving torque generated by the coil, and the higher the risk of position change, the amplitude of the excitation signal should not be too high. According to Figure 5, when the input signal amplitude is 2 V, the variation range of the mutual inductance voltage is approximately 10–120 mV, which is distinguishable already. Therefore, the input signal for the subsequent content is a sinusoidal signal with 50 Hz and 2 V amplitude.

There are three layers of coils in the stator, while the rotor teeth are only distributed in one single layer. In contrast to the 60° symmetrical periodicity of the coil mutual inductance generated by the rotor yawing, the mutual inductance voltage generated during the rotor pitch movement is only symmetrical but not periodic. In order to verify this analysis, an experiment is carried out by spinning the rotor for 60° from the initial position and recording the mutual inductance voltages of the fifth coil group as shown in Figure 6. While the rotor is pitching 3°–45° from the initial position, the mutual inductance voltages of the first group of coils are shown in Figure 7. From Figures 6 and 7, it can be seen that limited by the fact that the mutual inductance voltage caused by rotor yawing only has a monotonic variation range of 30°, each group of coils can detect that the angle range between rotor teeth and the coil axis is within ±30°.

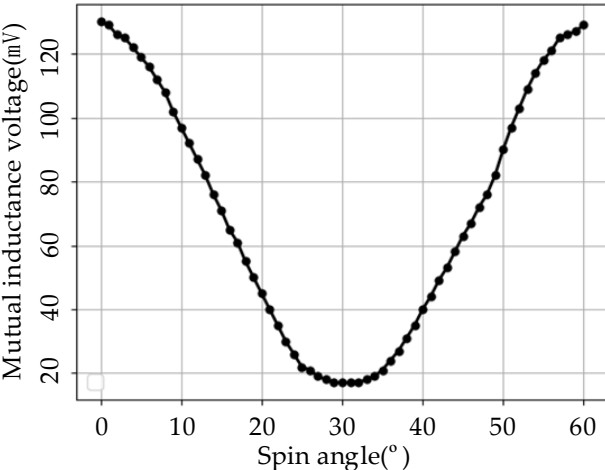

**Figure 6.** Mutual inductance voltage measurement value of group 5 coil at yaw range of 60°.

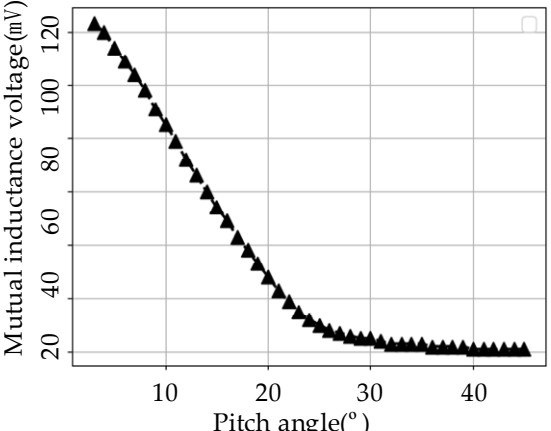

**Figure 7.** Mutual inductance voltage measurement value of group 1 coil in pitch range of 3°–60°.

After the measurement is completed, take the latitude and longitude position of the rotor pole relative to the coil axis on the spherical surface as the coordinate, mark the mutual inductance voltage value on a 2D plane, and draw a contour map of the mutual inductance voltage accordingly. The predicted value of the mutual inductance voltage when the coil is at this assumed position can be obtained by interpolating the contour map with the longitude and latitude of the assumed position point. This predicted value would be used to compare with the measured value to judge the rationality of the hypothetical position point.

For the same group of coils, the variation patterns of the mutual inductance voltage caused by rotor yawing and pitching are not completely consistent. The test results for the mutual inductance voltage generated in coil group number 5 with the rotor spinning and pitching 30°, respectively, from the initial position, are shown in Figure 8. Due to the difference in the magnetic circuit structure symmetry of the motor, the mutual inductance voltages generated at the yaw angle and pitch angle of the same size are slightly different, which makes the two curves in Figure 5 not coincide, and the contour lines of the mutual inductance voltage present an elliptical shape as shown in Figure 9. Moreover, due to the inconsistent magnetic circuit conditions caused by the difference in processing and installation of each group of coils, the details such as the size and spacing of mutual inductance voltage contour line ellipses measured by each group of coils are also inconsistent. This feature would make the interpolation processing of the mutual inductance voltage

contour map not only depend on the position point where the coil center is located, but the voltage gradient direction of this position point would also need to be considered.

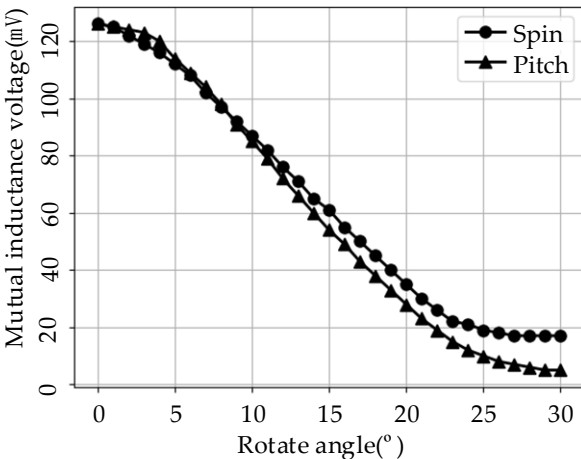

**Figure 8.** Mutual inductance voltage measurements for yaw and pitch motion in the 30° range of the layer coil.

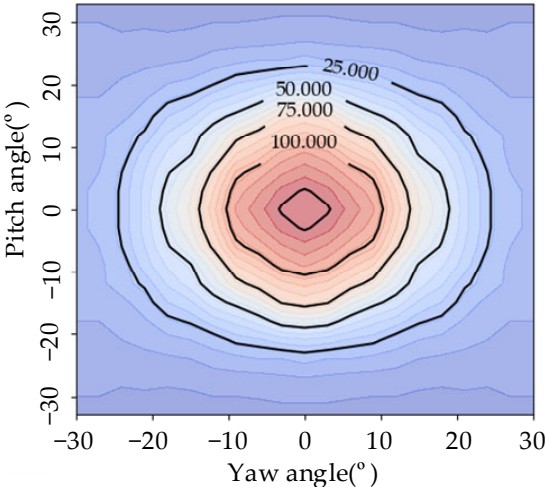

**Figure 9.** Contour diagram of mutual inductance voltage (mV) distribution.

### 3. Particle Swarm Optimization Algorithm for 3-DoF Attitude Recognition

#### 3.1. Identification Principle

According to the monotonous correspondence within a range of 30° between the mutual inductance voltage generated in the coil and rotor teeth misalignment angle, each group of coils can be regarded as a sensor for detecting the rotor teeth. Figure 10 shows the detection range of twelve coil groups of the upper, middle, and lower layers in the range of 0° to 180° east longitude of the stator, and the western half of the sphere is symmetrical. In the figure, the center of each coil's detection range is the installation position of the coil on the stator. Each ellipse represents the maximum detection range of a coil., and the filled circles represent the positions of the three rotor teeth located in the eastern hemisphere surface at a rotor random position. Theoretically, as long as any two of the three rotor teeth can be determined, the 3-DoF position angle of the rotor can be calculated. It is the simplest to calculate the spherical position of rotor teeth by the trilateration method [17,18], judging the position of rotor teeth by the distance from rotor tooth to three

stator coils with known positions. However, due to the number of rotor teeth in the prototype, the period of coil mutual inductance voltage is too short, which makes the detection range of each coil detector too small. As shown in Figure 10, the coincidence of the detection ranges of the third, fourth, and seventh coils is only a small piece of black zone, and the condition of trilateration is met only when the rotor teeth are located in this area. However, the area that meets this condition only accounts for a small part of the possible positions of the whole rotor tooth.

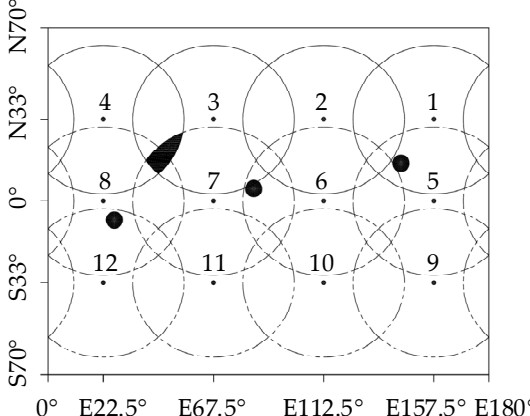

**Figure 10.** Distribution of the detectable range of each group of coils.

On the other hand, according to the contour map of the coil mutual inductance voltage obtained in advance, the theoretical value of the coil mutual inductance voltage can be conveniently obtained from the positional relationship between the rotor magnetic pole and coil axis. However, if the position of the rotor magnetic pole according to the value of the mutual inductance voltage is reversed, a position set represented by a contour line instead of a certain position value would be obtained. Therefore, it is impossible to directly calculate the rotor position from the mutual inductance voltage. Instead, it is necessary to constantly test whether a hypothetical rotor position can bring about the mutual inductance voltage that is consistent with the actual situation, that is, to invert the rotor position through the information of the mutual inductance voltage. If the assumed rotor 3-DoF position can bring the assumed values of the mutual inductance voltage of all coils close enough to the measured values, it can be considered that the assumed position of the rotor is approximately equal to the actual position. According to this train of thought, in the process of rotor position inversion, the measured values of the mutual inductance voltage of all coils are measured and recorded online on the one hand, assuming the 3-DoF position angle of the rotor on the other hand. Then, determine the assumed position of each rotor magnetic pole in combination with the design structure and obtain the assumed value of the mutual inductance voltage of each coil at this assumed position according to the known contour map of mutual inductance voltage. An intelligent optimization algorithm is used to optimize the assumed value of the rotor 3-DoF position angle, and the assumed value that minimizes the error between the predicted value and the measured value of all mutual inductance voltages is found as the final rotor position inversion result.

The specific operations are as follows: According to the relationship between the mutual inductance voltage of the stator coil and the included angle between stator and rotor, the particle swarm optimization algorithm is used to identify the three Euler angles $[\alpha^*, \beta^*, \gamma^*]^T$ of the rotor. The three-DoF rotation angle $[\alpha, \beta, \gamma]^T$ of the rotor is taken as the optimization variable, and the error between the measured value $U^* = [u_i^*]^T, i = 1 \sim 12$ and the theoretical value $U = [u_i]^T$ of the mutual inductance voltage of the 12 sets of stator coils is taken as the fitness function $fitness(\alpha, \beta, \gamma)$, and the minimum fitness function

$min\ [f(\alpha,\beta,\gamma)]$ is taken as the optimization objective. There are two data preparations that should be made upfront:

(1) According to the method introduced in Section 2.2, the mutual inductance voltage values $u_{iyaw} = f(\theta_{spin})$ and $u_{ipitch} = f(\theta_{pithc})$, corresponding to the rotor yaw and pitch motion within 0°–30° for each group of coils, are measured in advance as the interpolation basis for calculating the theoretical value of the mutual inductance voltage. Among them, $i$ is the number of 12 groups of stator coils; $\theta_{iyaw}$, $\theta_{ipitch} \in (0^{\circ}{\sim}30^{\circ})$, $u_{iyaw}$, and $u_{ipitch}$ are the mutual inductance voltages measured by coil i during rotor yaw and pitch, respectively.

(2) According to the basic attributes of the ellipse, calculate the spherical longitude and latitude $(X_{ci1}, Y_{ci1}),(X_{ci2}, Y_{ci2})$ with $i = 1{\sim}12$ of the contour line of the mutual inductance voltage of each coil, and the spherical latitude and longitude of the two focal points of the ellipse.

When on-line identification of the three-DoF angle is carried out, firstly, the actual mutual inductance voltage numerical vector $[\alpha^*,\beta^*,\gamma^*]^T$ of each group of coils is obtained by test in the state of ball machine to be measured $U^* = [u_i^*]^T, i = 1{\sim}12$. After that, the initial solution set of the particle swarm optimization algorithm is adopted and iterated according to the fitness value. The specific fitness calculation process is as follows.

Step 1: According to the input three angles $[\alpha,\beta,\gamma]^T$, the spherical longitude and latitude positions $[X_{ti}, Y_{ti}]^T, (i = 1{\sim}6)$ of the six rotor teeth in this state are calculated by using the three-dimensional space geometric relationship.

Step 2: It is believed that the mutual inductance of a group of coils is mainly determined by the pair of rotor teeth closest to the pair of coils. Therefore, for 12 coils in the Eastern Hemisphere stator, according to the properties of the ellipse, the sum $\left[d_{citj}\right]^T, i = 1{\sim}12, j = 1{\sim}6$ of the distances between the six rotor teeth and the two focal points of the ellipse of the mutual inductance voltage contour line of each coil is calculated, the minimum value $a_i = \min[d_{citj}]$ is taken, and the mutual inductance voltage of the coil is determined by this distance.

Step 3: Calculate the length $a_i$ of the short axis according to half of the long axis of the ellipse corresponding to $2c_i = \|Y_{ci1} - Y_{ci2}\|$ the mutual inductance voltage contour line and the focal length of the ellipse $b_i = \sqrt{a_i^2 + c_i^2}$. Use $a_i$ in $u_{ispin}$ and $b_i$ in $u_{ipitch}$ test data interpolated and averaged as the theoretical value $u_i$ of the mutual inductance voltage to be generated by this coil.

Step 4: After obtaining the calculated value $U = [u_i]^T$ of the mutual inductance voltage of 12 groups of stator coils, calculate the root mean square error between the measured value and the calculated value as the fitness value $fitness(\alpha,\beta,\gamma) = \sqrt{\frac{1}{12} \sum_{i=1}^{12}(u_i^* - u_i)^2}$ of individual algorithm population.

### 3.2. Experimental Verification

In order to verify the effectiveness of the sensorless detection method proposed in this paper, experimental verification is carried out by using artificially preset posture points and comparing them with the detection results of contact sensors. According to the readings of rotary encoders installed on three rotating shafts on the 3-DoF contact position detection bench as shown in Figure 3, it is considered that the rotor of the spherical motor is adjusted to a given position. Then, the mutual inductance voltage values of all twelve groups of coils are detected on a tour inspection using the detection circuits shown in Figure 4. These voltage values are put into the position recognition program for algorithmic recognition. The recognition results are compared with the actual readings of the contact sensor. The main parameters of the algorithm are set as follows: swarm size is 200, iteration algebra is 50, and fitness function is described in the previous section.

Two motion tracks are designed for the test. The first is to fix the roll angle of the rotor around the X-axis at −10°, the pitch angle around the Y-axis increases from −10° to

10° in steps of 2°, and the yaw angle around the Z-axis synchronously decreases from 20° to −20° in step of 4°. The experimental results are shown in Figure 11. The dashed line is the actual value of the position angle obtained by the contact position sensor, and the dotted line is the detection value of the position angle obtained by inversion according to the mutual inductance voltage.

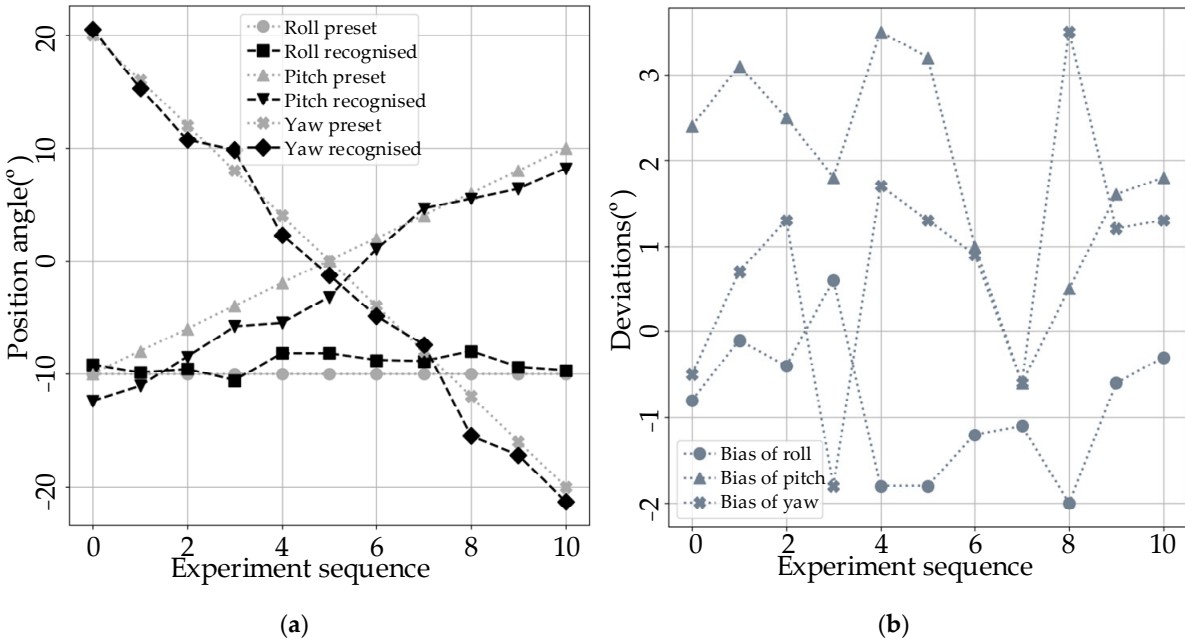

(**a**)                    (**b**)

**Figure 11.** Result of experiment 1: (**a**) experimental result, (**b**) error distribution.

The second experimental trajectory is as follows: all three degrees of freedom rotate synchronously in a step of 3°, in which the pitch angle around the Y axis and the yaw angle around the Z axis decrease from 15° to −15°, and the pitch angle around the Y axis increases from −15° to 15°. The experimental results are shown in Figure 12.

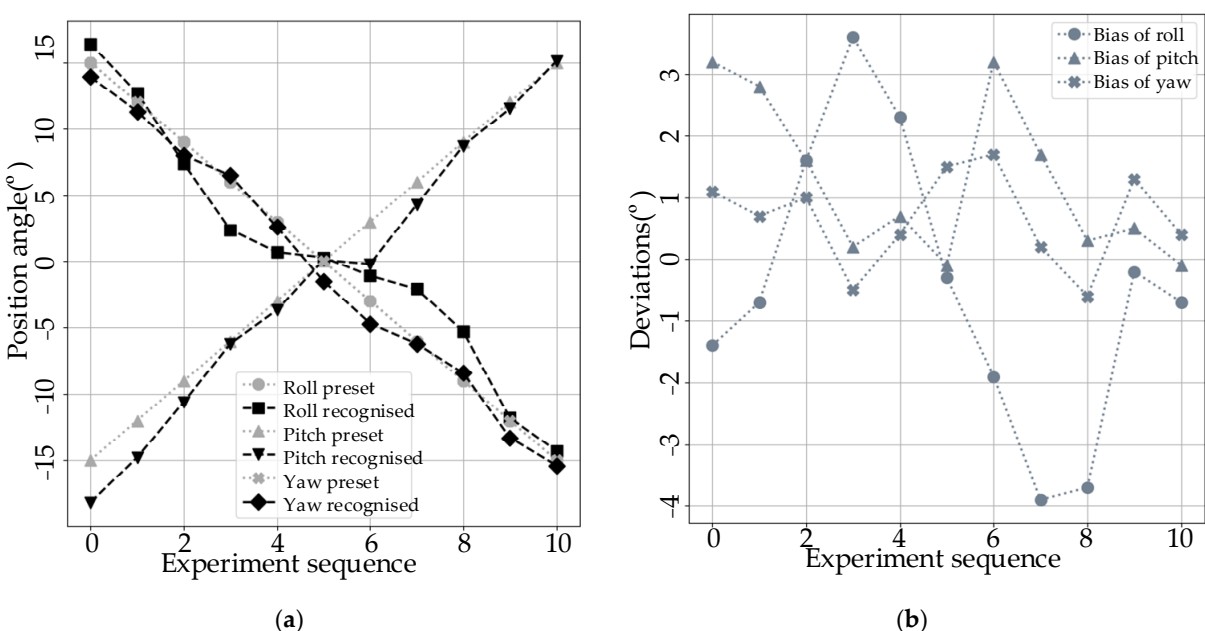

(**a**)                    (**b**)

**Figure 12.** Result of experiment 2: (**a**) experimental result, (**b**) error distribution.

From the comparison between the set value and the detected value of the 3-DoF position angle in Figures 11 and 12 and Tables 1 and 2, it can be seen that the proposed sensorless detection method has a good on-line detection effect; the maximum detection error is within ±4°, and the population standard deviation is within 1.8°. It can be used for replacing the position detection method with sensors.

**Table 1.** Preset; detection values; and deviations of roll, pitch, and yaw of experiment 1 (°).

| Test series | 1 | 2 | 3 | 4 | 5 | 6 | 7 | 8 | 9 | 10 | 11 |
|---|---|---|---|---|---|---|---|---|---|---|---|
| roll preset | −10 | −10 | −10 | −10 | −10 | −10 | −10 | −10 | −10 | −10 | −10 |
| roll recognized | −9.2 | −9.9 | −9.6 | −10.6 | −8.2 | −8.2 | −8.8 | −8.9 | −8 | −9.4 | −9.7 |
| pitch preset | −10 | −8 | −6 | −4 | −2 | 0 | 2 | 4 | 6 | 8 | 10 |
| pitch recognized | −12.4 | −11.1 | −8.5 | −5.8 | −5.5 | −3.2 | 1 | 4.6 | 5.5 | 6.4 | 8.2 |
| yaw preset | 20 | 16 | 12 | 8 | 4 | 0 | −4 | −8 | −12 | −16 | −20 |
| yaw recognized | 20.5 | 15.3 | 10.7 | 9.8 | 2.3 | −1.3 | −4.9 | −7.4 | −15.5 | −17.2 | −21.3 |
| bias of roll | −0.8 | −0.1 | −0.4 | 0.6 | −1.8 | −1.8 | −1.2 | −1.1 | −2 | −0.6 | −0.3 |
| bias of pitch | 2.4 | 3.1 | 2.5 | 1.8 | 3.5 | 3.2 | 1 | −0.6 | 0.5 | 1.6 | 1.8 |
| bias of yaw | −0.5 | 0.7 | 1.3 | −1.8 | 1.7 | 1.3 | 0.9 | −0.6 | 3.5 | 1.2 | 1.3 |
| population standard deviation | | | | | | | 1.7 | | | | |

**Table 2.** Preset; detection values; and deviations of roll, pitch, and yaw of experiment 2 (°).

| Test series | 1 | 2 | 3 | 4 | 5 | 6 | 7 | 8 | 9 | 10 | 11 |
|---|---|---|---|---|---|---|---|---|---|---|---|
| roll preset | 15 | 12 | 9 | 6 | 3 | 0 | −3 | −6 | −9 | −12 | −15 |
| roll recognized | 16.4 | 12.7 | 7.4 | 2.4 | 0.7 | 0.3 | −1.1 | −2.1 | −5.3 | −11.8 | −14.3 |
| pitch preset | −15 | −12 | −9 | −6 | −3 | 0 | 3 | 6 | 9 | 12 | 15 |
| pitch recognized | −18.2 | −14.8 | −10.6 | −6.2 | −3.7 | 0.1 | −0.2 | 4.3 | 8.7 | 11.5 | 15.1 |
| yaw preset | 15 | 12 | 9 | 6 | 3 | 0 | −3 | −6 | −9 | −12 | −15 |
| yaw recognized | 13.9 | 11.3 | 8 | 6.5 | 2.6 | −1.5 | −4.7 | −6.2 | −8.4 | −13.3 | −15.4 |
| bias of roll | −1.4 | −0.7 | 1.6 | 3.6 | 2.3 | −0.3 | −1.9 | −3.9 | −3.7 | −0.2 | −0.7 |
| bias of pitch | 3.2 | 2.8 | 1.6 | 0.2 | 0.7 | −0.1 | 3.2 | 1.7 | 0.3 | 0.5 | −0.1 |
| bias of yaw | 1.1 | 0.7 | 1 | −0.5 | 0.4 | 1.5 | 1.7 | 0.2 | −0.6 | 1.3 | 0.4 |
| population standard deviation | | | | | | | 1.75 | | | | |

The main causes of experimental biases are as follows:

1.  The bias brought from inconsistent magnetic circuits caused by the manufacturing process and assembly accuracy of the spherical motor. In the process of testing and drawing the contour lines of the mutual inductance voltage, it is found that the contour lines obtained by selecting different rotor pole as regulating object have certain differences. In this regard, this paper takes the median processing of the mutual inductance voltage contour lines obtained from the different rotor pole tests. The difference between this median and the actual measurement value would be reflected in the bias of the final result. This part of bias can be reduced by improving the manufacturing process of the motor and improving the consistency of the magnetic circuit.
2.  The bias brought by the algorithm termination condition. In order to ensure the output speed of the algorithm, a coercive termination condition according to evolutionary algebra is set. This may lead to some cases in which the fitness of the output results is not high enough. Therefore, it is necessary to further improve the efficiency of the algorithm, reduce the amount of computation, and improve the convergence speed.
3.  The deviation of the preset position angle during the experiment. During the verification experiment, the rotor is first manually rotated to the preset position according

to the reading of the contact sensor. Due to the need to meet the position requirements in three directions at the same time in the manual process, certain deviations would inevitably occur. In addition, in the process of measurement, the rotor would be slightly displaced due to the gravity of the additional mechanism and other factors, which would also lead to the deviation between the measured angle and the preset angle.

## 4. Conclusions

Aiming at the main problems existing in the 3-DoF position detection in the current research and taking lessons from the basic principle of sensorless position detection of a traditional rotating electrical motor, the reluctance spherical motor with a nonlinear magnetic circuit and iron core is taken as the object to make full use of its doubly salient characteristics and study the position of the sensorless detection method for the spherical motor. Starting from the influence of the structure of the spherical motor on the numerical and spatial variation range of the stator coil inductance and using a mathematical analysis, a computer simulation, and experimental verification methods, a sensorless position detection method is realized to detect the rotor's spatial position through the coil mutual inductance voltages. The experimental results show that the detection method has a good on-line detection effect, and the population standard deviation is within 1.8° Therefore, the developed technique can be used for replacing the position detection method with sensors. This paper aims to achieve a breakthrough in the key technology of 3-DoF position detection of spherical motors through systematic research and promote the development and the practical process of multi-DoF motors. Following the research presented in this paper, we plan to further improve the detection accuracy while simplifying the detection circuit and algorithm to speed up the detection speed, thereby improving the applicability of the proposed sensorless detection method.

In addition to realizing sensorless 3-DoF position detection, the following conclusions can be drawn by summarizing the research process of this paper:

1.  Sensorless position detection is often carried out by using the salient pole effect or nonlinear saturation characteristics of the motor. In order to avoid excessive reluctance torque, permanent magnet spherical motors usually adopt a coreless design; thus, there are no convex polarity or nonlinear saturation characteristics that can be utilized. Compared with permanent magnet spherical motors, it is easier for the reluctance spherical motor to realize sensorless position detection.
2.  The design structure of the spherical motor has great influence on the sensorless position detection effect. For example, the periodicity of the rotor structure corresponds to the periodicity of the detection result. From the perspective of the detection range, the number of rotor magnetic poles should be as small as possible. Therefore, whether from the perspective of subsequent drive control or position detection, the influence should be considered at the early stage of motor design.

**Author Contributions:** Conceptualization, J.X. and R.Z.; funding acquisition, Q.W., R.Z. and L.J.; investigation, L.J.; methodology, J.X.; project administration, Q.W. and G.L.; software, R.Z.; validation, J.X. and S.Z.; writing—original draft, J.X.; writing—review and editing, Y.W. All authors have read and agreed to the published version of the manuscript.

**Funding:** This research was funded by the key project of the China National Natural Science Foundation, grant number 51637001 and was also funded by Anhui Natural Science Fund Project, grant number 1908085QE236, 1908085ME168 and 2008085ME156.

**Conflicts of Interest:** The authors declare no conflict of interest.

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
