# Peer review of "Sensorless Posture Detection of Reluctance Spherical Motor Based on Mutual Inductance Voltage"

_applsci, doi:10.3390/app11083515_

Round 1

Reviewer 1 Report

  • Good abstract lacks any pithy statement of results worded in broadest terminology. Good examples include percentage difference between experimental and predicted values (e.g. means and standard deviations of signals plotted in figures 11 and 12).
  • Please explain to the reader some level of detail to be sought in each respective reference in the triple citation [2-4].
  • Please explain to the reader some level of detail to be sought in each respective reference in the triple citation [5-7].
  • Please explain to the reader some level of detail to be sought in each respective reference in the double citation [8-9].
  • Please explain to the reader some level of detail to be sought in each respective reference in the double citation [9-10].
  • Please explain to the reader some level of detail to be sought in each respective reference in the quintuple citation [11-15] in line 88.
  • Please explain to the reader some level of detail to be sought in each respective reference in the triple citation [16-18] in line 247.
  • Figure 3 is marginally useful due to small text size.
  • Figure 4a transmits no information to readers due to illegibility. Figure 4b is marginally useful due to small text size.
  • Figure 5 a,b,c are marginally useful due to small text size.
  • Figures 6-8 are marginally useful due to small text size and slight blur.
  • Figure 9 is essentially useless to readers with printed copies of the manuscript, since the abscissa and ordinate labels are illegible.
  • Figures 11 and 12 are marginally useful, since the reader cannot distinguish identical line styles when the manuscript is printed (especially in black and white copies). Please consider modifying line thicknesses as an additional feature to distinguish lines of identical line styles.
  • MAJOR STRENGTH: the experimental validation increases the manuscript’s quality. MAJOR WEAKNESS: results were merely presented qualitatively (readers are to note differences in plotted lines of data), while quantitative results are immediately available, e.g. means and standard deviations of the differences plotted.  The review strongly recommends adding a table of quantitative results, and then also amplify the conclusions and abstract with a single sentence describing the quantitative comparisons.
  • Please consider amplifying the conclusions section with statements of future research, so the readers can follow the chronology of development by the authors. The reviewer recommends additional comparison with deterministic artificial intelligence recently applied to DC motors which seems comparable to the application here on spherical motor mutual induction voltage: Sands, T. Control of DC Motors to Guide Unmanned Underwater Vehicles. Sci. 2021, 11(5), 2144.

Reviewer 2 Report

Thanks for your interesting paper. However i have some comments to modify the presentation of your paper

1- It will be interesting if you added a 3-D picture for the motor or pictures from different sides.

2- Equations format is not good. A modification is mandatory.

3- modify figure 3,4 and 5.

4- In line 147 page 5, the variable UA seem to be added as picture please use word equation to insert variables. Modify this comments for the whole paper.

5- Font of axis number in figure 6 and 7 and also for the next figures must be magnified.

6- There is no space between figures and paragraph also format of figures title aren't good.

7- the finding of this paper is not clearly presented in conclusion part.

8- Reference is too old. You have to update your reference to 2019,2020 and 2021
